# An Energy-Based Assessment of Expected Benefits for V2H Charging Systems through a Dedicated Dynamic Simulation and Optimization Tool

Carlo Villante *[iD], Stefano Ranieri *, Francesco Duronio [iD], Angelo De Vita and Michele Anatone

Industrial Engineering, Information and Economics Department, University of L'Aquila, 67040 L'Aquila, Italy; francesco.duronio@univaq.it (F.D.); angelo.devita@univaq.it (A.D.V.); michele.anatone@univaq.it (M.A.)
* Correspondence: carlo.villante@univaq.it (C.V.); stefano.ranieri@graduate.univaq.it (S.R.)

**Abstract:** Electricity from renewable energy sources represents the most promising way to decarbonize energy systems. A grid connection of car Electricity Storage Systems (ESSs) represents an opportunity to tackle issues regarding electricity production non-programmability, only if sufficiently smart bi-directional Vehicle to Grid technologies (V2G) are widely implemented. Fully Bi-directional grid capabilities are still poor and must be increased, both physically and in terms of management and billing possibilities (in the so-called smart-grid paradigm). However, some V2G technologies may be already implemented in smaller individual contexts: so-called Vehicle to Home, V2H technologies. Starting from these considerations, within the frame of an Italian publicly funded research project, the authors categorized and described many possible application contexts and developed an open-source dynamic simulation (fully available under request for the scientific community) to identify most promising conditions. To this aim, they also synthetized and tested an effective energy optimization algorithm which will soon be implemented on a prototypal wireless V2H device, built by ENEA in cooperation with Cassino University, in Italy. The performances of the system were assessed evaluating electricity auto-consumption and home auto-feeding ratios. Simulations show that very relevant performances can be obtained, up to the values 69% for electricity auto-consumption and 82% of home auto-feeding.

**Keywords:** battery electric vehicle; vehicle-to-grid; vehicle-to-home; simulation tool; optimal dimensioning

## 1. Introduction

Currently, transport systems and their related service infrastructures are mainly based on internal combustion engines and fossil sources. The transport sector is the most challenging to be decarbonize and is currently responsible for 37% of worldwide direct CO2 emissions from the end-user sector (2020 annual emission reaching 7.2 GtCO2 [1]). Moreover, transportation is also largely the most relevant sector contributing to air pollutants emissions (NOx, CO, HC, PM) in urban environments, which, in turn, were estimated to be responsible for 4.2 million of premature deaths worldwide in 2016 [2]. This certainly renders fossil fuels no-longer sustainable, especially for urban transport applications, due to both the damages to people and the environment, and to their depletion [3].

It therefore becomes essential to rapidly increase transportation sustainability, still satisfying user expectations and transportation needs as far as possible. Certainly, the most attractive among the emerging alternatives is electric mobility: electricity is used in place of liquid fuels as energy vectors, substantially contributing to pollution reduction in urban contexts, and simultaneously finding new utilization space for electricity produced by renewable energy sources (RES) (e.g., wind, hydro, and solar energy). The sustainability of our whole energy system may be strongly increased, thanks to the widespread availability and inherently pollutant-free nature of RES. According to this transition paradigm, electric mobility shares are currently rapidly increasing, already reaching over 10 million electric

vehicles (EV) worldwide, including Battery Electric Vehicles (BEVs), Plug-in Hybrid Vehicles (PHEVs), and Fuel Cell Electric Vehicles (FCEVs), namely over 8 times more compared to 2015 (1.2 million electric cars). However, despite its rapid growth, electric vehicles in 2020 still account for only 1% of the overall circulating stock in the light duty (LD) transport sector. According to International Energy Agency (IEA) those shares will rapidly grow in the near future and will reach among 7.5% and 12.8% by 2030, as shown in Figure 1, depending on the projection scenario [4]. Indeed, in the Stated Policies Scenario (STEPS—namely a situation that reflects all existing policies and targets that have been legislated or announced by governments around the world) the estimated average annual growth rate of EV stock will be 30%, reaching almost 140 million light duty vehicles (LDVs) sold by 2030, while in the so-defined Sustainable Development Scenario (SDS), over 220 million electric LDVs are projected to be circulating by 2030, corresponding to an almost 12.8% stock share. SDS is a more ambitious scenario, in line with EV30@30 signatories, a campaign launched at the 8th Clean Energy Ministerial (CEM) in June 2017 and supported by 15 member countries and several international companies and organizations, whose objective include achieving a 30% sales share in 2030 for light duty EVs [5].

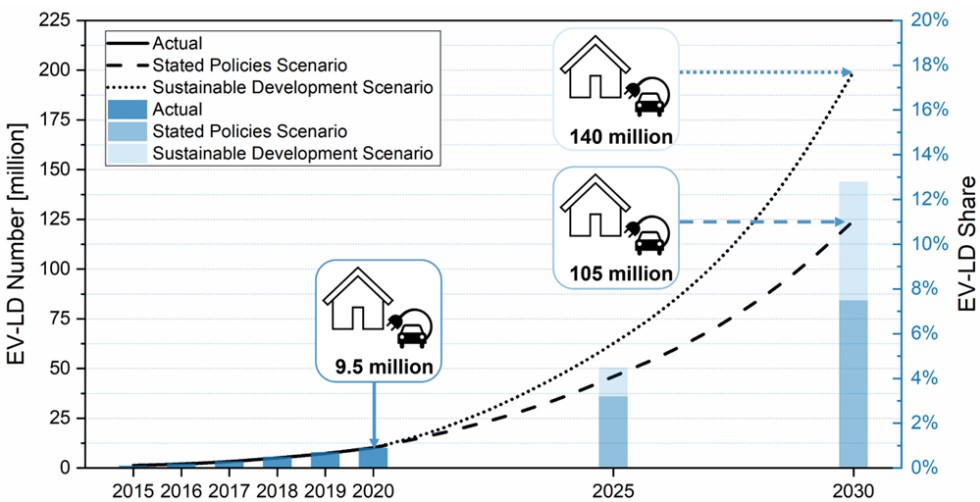

**Figure 1.** Electric Vehicle historical and prospective worldwide data. Line chart: number of electric light duty vehicles. Bar chart: share of electric light duty vehicle stock on total transport sector. Outlined box: number of private electric chargers.

Several factors will contribute to the rapid increase of the EVs market, primarily the reduction of future purchase costs (also thanks to fiscal incentives provided by governments) and the rapid increment of battery performances, weight and economic convenience [6–8].

However, a fast and massive transition to electric mobility will present several issues and unpublished problems to be faced. Those include the very demanding conditions in terms of electrical power required from the electricity grid to charge high numbers of vehicles Electric Storage Systems (ESSs) connected in very restricted areas and for limited time periods. However, problems arise also from an energy grid balance perspective, being that electric energy production from RES is mostly unpredicted and non-programmable, making it impossible to match instantaneously the power production and demand (which is a pre-requisite for power grid control and power quality insurance) [9].

All these problems call for the need of a great amount (many hundreds of GWh capacity on a national level) of ESSs to be connected to the grid and usable for grid balancing and grid services purposes.

Therefore, the possibility arises of interfacing vehicles own ESSs with the electric grid smartly and bi-directionally, through so-called Vehicle-to-Grid (V2G) devices. This would make it possible, at least partially and locally, to cope with the ever-increasing demand for energy and power and their dynamic variations. Indeed, one of the first advantages of V2G

technologies would be to provide ancillary services, which include stabilization of the grid voltage by coping with variations in demand, providing reactive power from the vehicle, and frequency stabilization, providing active power [10].

One of the major issues hindering the spread of V2G systems concerns the contractual front between customers and suppliers, in particular the management of a potentially high overall accumulation but of a continuously variable size and widespread ownership [11]. Most V2G technologies do not only require a local application because the electric grid may not be able to cope with their high requirements in terms of dynamic power requests. Some of them, however, may be already implemented on smaller scales, while vehicle ESSs are connected with individual homes (Vehicle-to-Home, V2H), as shown in Figure 2, or small/medium buildings (Vehicle-to-Building, V2B) charging points [12,13]. Both V2H and V2B systems do not require relevant modifications neither from an urbanistic nor an electric grid point of view. In addition, the management of these charging systems is much simpler than that of V2G ones, due to the type of connection, namely private and dedicated to the interface with known vehicles and in a well-defined contractual context [14].

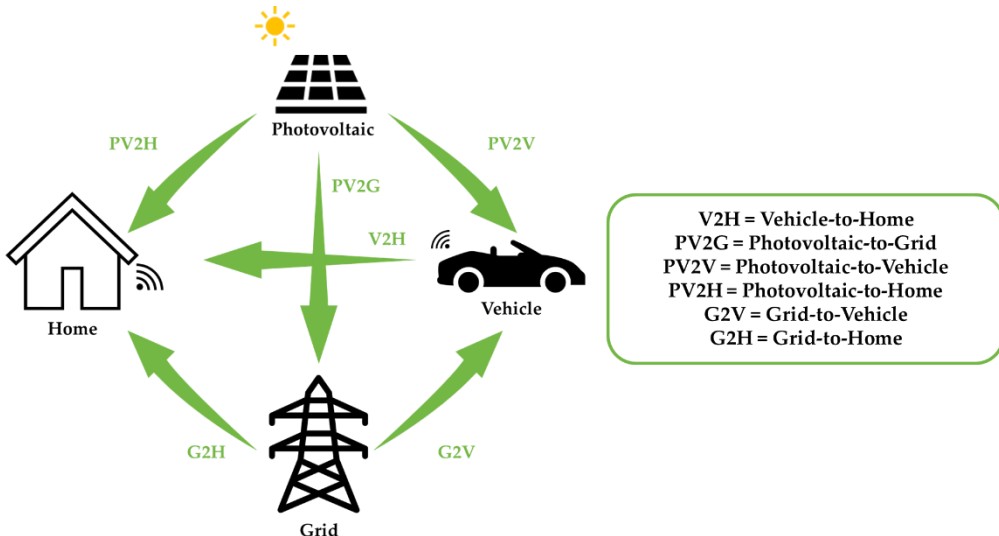

**Figure 2.** Illustration of V2H concept with main power flows.

Finally, V2H and V2B technologies will benefit from the continuous increase in the number of private charging stations, which will continue to grow in the coming years, as estimated by IEA [4] and shown in Figure 1. However, these applications have narrower margins of effectiveness and applicability with respect to other V2G applications, because ESSs tend to be grid connected in periods of low local electricity consumption, and poor electricity production from local renewable sources (e.g., photovoltaic PV devices).

Moreover, these technologies may not be interesting for all categories of users, as their potential benefit is strictly connected with the expected user's behavior both from a mobility (frequency, timespan, and amplitude of electric vehicle usage) and a domestic appliances usage perspective. Lastly, a crucial role is played by the type of apartment considered, being that those technologies are much more interesting only for those new-concept houses and buildings equipped with Photovoltaic plants (PV), and with electric HVAC technologies aimed at maximizing RES usage (invertible heat pumps both for winter heating (HP), and for summer cooling (AC)) [15]. For these applications, bidirectional V2H systems can potentially lead to significant economic benefits, when compared to stationary storage batteries [16].

The development of optimization systems for energy and economic management of those V2H and V2B systems therefore becomes crucial. Several approaches have been recently proposed for the energy optimization of V2G systems [17–22]. An overview of the most recent energy management systems for V2G application can be found in the work of

Ouramdane et al. [17]. For the sake of brevity, in the following few articles a summary of the most recent state-of-art V2H energy optimization systems are presented.

Slama [18] proposed a scheduling home energy management system for improving the performance of V2H systems, considering both travel times for the electric vehicles and climatic conditions. The proposed algorithm led to perform correct system operations and satisfy load demand, but it has not yet been implemented on a real-time system.

Mohammad et al. [19] used a heuristic gray wolf optimization algorithm for solving the non-linear multi-objective optimization problem of a home energy management system (HEMS) integrating RES, ESS, and several domestic appliances. The multi-objective function was represented by the weighted sum of two separated objective functions, namely the total energy cost and the ratio of peak demand to the daily average demand of all requests. Capabilities of the proposed system to improve V2H performances were demonstrated. However, the proposed HEMS was not capable of configuring household appliances in a way that they respond to real-time changes. For fully exploiting the capabilities of the proposed HEMS system its integration with artificial neural networks (ANN) would be necessary.

Ouramdane et al. [20] presented an optimal sizing and energy management algorithm for V2H systems, based on the Interior-point algorithm. They provided a comparison analysis between two case studies, one without the V2H system and the other with the V2H system sized and managed through the proposed algorithm. In their model, the authors considered the availability of the vehicle's ESS only during evening, night, and early morning hours. The results of the simulations conducted under real weather data for two different French cities demonstrated the capability of the V2H system for maximizing home autonomy and reducing the energy issued from the grid for both cities.

Wang et al. [21] presented an optimal energy management strategy for controlling the bidirectional energy flows in V2H systems based on the Jaya algorithm according to several constraints, such as electricity price, specific daily load profile, photovoltaic generation profile and electric vehicle travel distance. The simulation results highlighted the algorithm's ability to reduce daily electricity costs in V2H systems. However, the work does not provide a comparison between different electrical behaviors of the users.

Irtija et al. [22] proposed a novel demand-response management framework for residential smart grids, based on the principles of labor economics. After identifying the types of household "prosumers" (producers + consumers) and their characteristics in terms of electricity generation and consumption, they formulated and solved a contract-theoretic optimization problem with the aim of maximizing the economic profit of both prosumers and electricity market.

The above literature clearly highlights advantages and drawbacks of smart grid energy management systems. Depending on the optimization methodology used, some of the key issues related with functional algorithms are connected to the complexity and the required computational times of numerical algorithms, which, in turn, make their implementation in real-time devices particularly challenging.

Taking the lead from these considerations and previous experiences by other research groups, the authors here present the main results of a modeling activity funded by the Italian Ministry of Ecologic Transition (MITE) through its RSE (Electric System Research) funding scheme. All the results shown, SW tools and HW demonstrators developed under this funding scheme are public and may be freely used by other research groups for individual and/or cooperating activities. In particular, a mathematical tool was developed and is here presented which is able to simulate V2H and V2B systems. The SW tool was developed within a broader collaboration between the University of L'Aquila, the Cassino University and the Italian National Agency for New Technologies, Energy and Sustainable Economic Development (ENEA), which includes the realization and testing of a wireless V2H prototype.

The main objective of the dynamic simulation which is here presented was the development of an open-access SW, used to simulate many possible application contexts,

identify the most promising among them, and to synthesize and test V2H systems control algorithms, which should be sufficiently simple to be easily installed and real-time implemented on a variety of residential devices, and not being dependent on relevant communications with vehicular ESS Battery Management System (BMS). The algorithm here proposed, in particular, was also implemented on a SpeedGoat Real-time machine and is currently under testing on a wireless V2H prototype realized by ENEA and Cassino University. All this activity will be soon object of a further publication by the research group.

The proposed SW tool was first used to simulate several possible users and contexts identifying the most promising conditions for V2H devices. To this aim, the authors also defined and calculated two simple V2H energy performance indicators, which will be described in the following.

The approach proposed was chosen to be replicable and modular also permitting its generalization and application to small/medium building contexts, in so-called V2B. This activity is currently under development and will be soon published in a further paper by the same authors.

## 2. Method

### 2.1. Classification of V2H End Users

The possible end users for the V2H system were classified in the following 5 categories, depending on the daily inhabited hours of the house:

1.  "Office-Workers" (**OW**): this category includes all those users that make use of the home in a prevalent pattern of use similar to that of an office worker. In this case it is foreseeable that the house will be not inhabited between 8am and 5pm of every working day. Therefore, an electric car could not be connected to the home electrical grid during the same time.
2.  "Smart-Workers" (**SW**): this category includes all those users that make use of the home in a prevalent pattern similar to that of a smart worker. In this case it is foreseeable that the house will be inhabited almost continuously. Therefore, an electric car could be connected to the network for almost the entire day.
3.  "Morning Shift Worker" (**MW**): this category includes all those users that make use of the home in a prevalent pattern similar to that of a morning shift worker. In this case it is foreseeable that the house will be not inhabited between 6am and 2pm of every working day. Therefore, an electric car could not be connected to the home electrical grid during the same time.
4.  "Afternoon Shift Worker" (**AW**): this category includes all those users that make use of the home in a prevalent pattern similar to that of an afternoon shift worker. In this case it is foreseeable that the house will be not inhabited between 2pm and 10pm of every working day. Therefore, an electric car could not be connected to the home electrical grid during the same time.
5.  "Nocturne Shift Worker" (**NW**): this category includes all those users that make use of the home in a prevalent pattern similar to that of a night shift worker. In this case it is foreseeable that the house will be not inhabited between 10pm and 8am of every working day. Therefore, an electric car could not be connected to the home electrical grid during the same time.

The described behaviors are valid during the weekdays, while, on holidays and weekends, they tend to be all similar to each other. For the purposes of this work, it was assumed that users leave their homes mainly on the days before holidays, while they tend to stay at home on holidays. By combining this last hypothesis with the characteristics for the individual end users, it is possible to obtain a hypothetical weekly plan for the connection of electric cars to the home recharging stations, as shown in Figure 3.

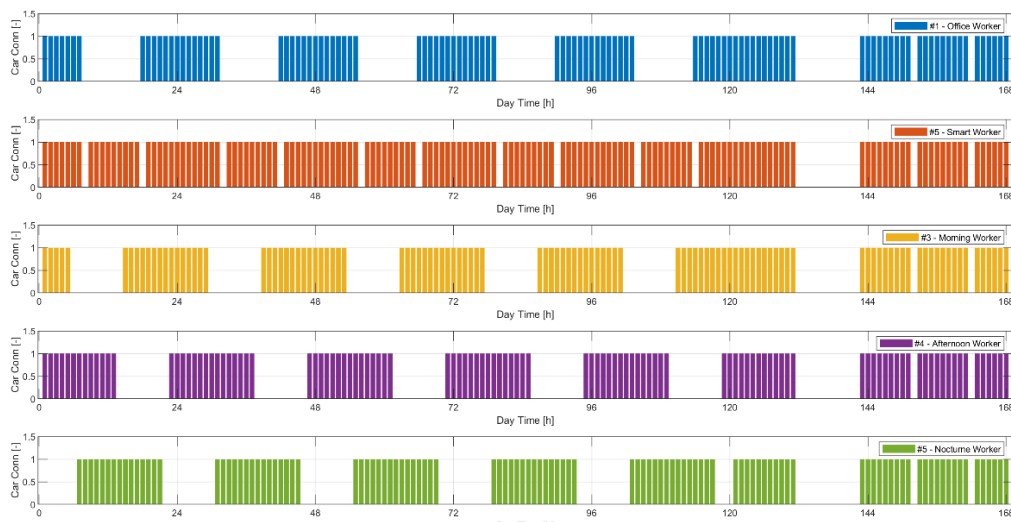

**Figure 3.** Weekly connection time of the electric vehicle to the home recharging equipment for the different end users.

### 2.2. Characterization of "Electrical Behaviors" for Each End User

An expected "electrical behavior" was associated for each end user, represented by the sum of electricity consumption and production attributable to it. The domestic electrical consumptions were distinguished in:

- Consumptions related to the presence of users at home, such as household appliances and electrical/electronic devices, including the home recharging station.
- Consumptions independent of the occupation of the home itself, including electricity related to the use of electrical heating, ventilation, and air conditioning (HVAC) systems.

The description of how the daily loads were calculated can be found in [23–25]. The sum of these two types of consumptions allowed the obtainment of an annual electricity consumption plan for each end user. Figure 4 shows this plan, detailed for 4 typical weeks in the different seasons of the year depending on the presence of heating and refrigeration loads.

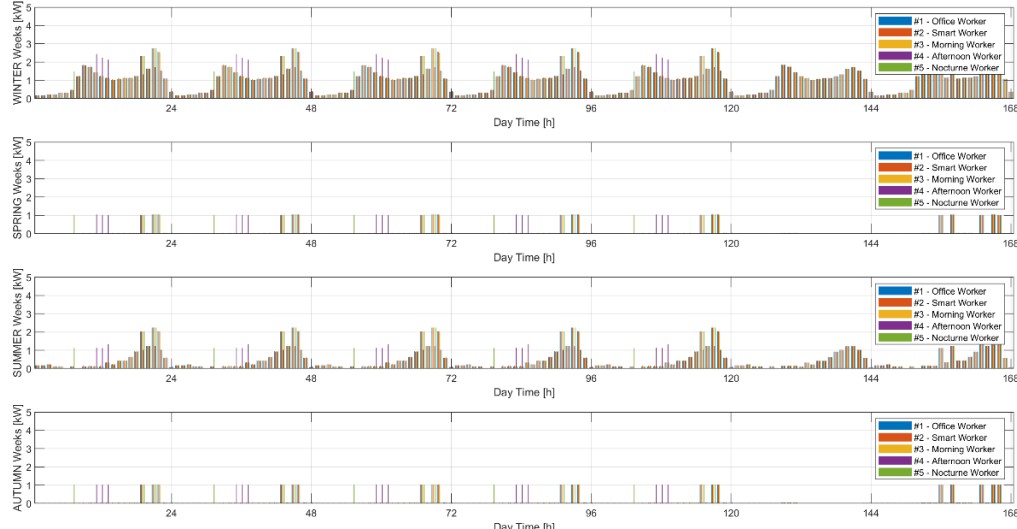

**Figure 4.** Total weekly electrical power consumption for the various seasons and end users.

Regarding the home energy production, the installation of a typical residential photovoltaic system was assumed. The energy that can be produced by a PV system clearly depends on the climatic conditions of the site where the house is located.

Figure 5 shows an estimate of the power that can be produced by a residential PV system during a typical year in a representative location of the average Italian context (1450 kWh/year of energy production for 1 kWp of power installed), as assumed in the present work. PV plant extension (about 40 m$^2$) and Nominal power production (4.5 kW) were chosen to be compatible with those of a common user contract in Italy (3 to 6 kW), and with a roof surface which may be available both in a single home and/or in a small/medium residential building.

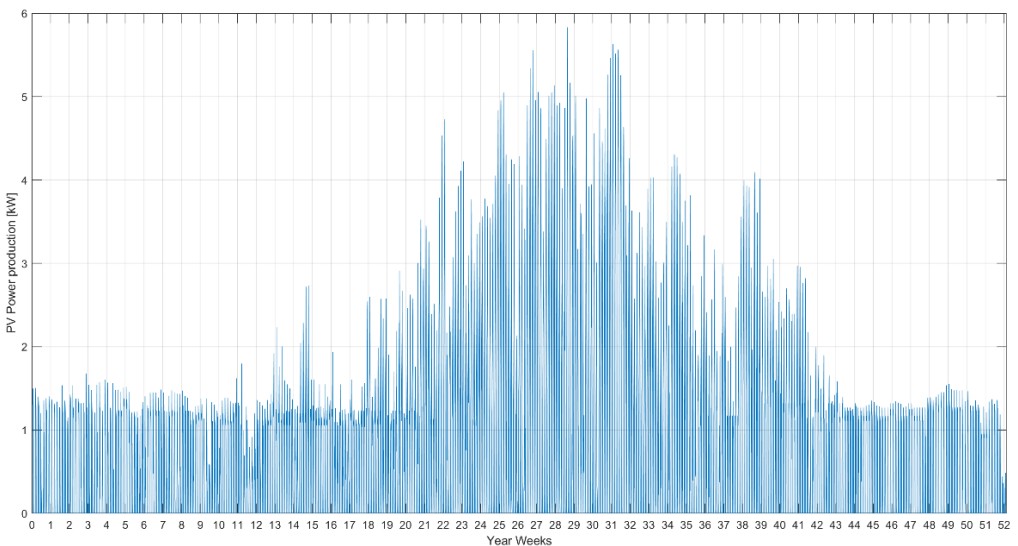

**Figure 5.** Estimation of power that can be produced by a residential PV system during the year, in the average Italian context.

Figure 6 shows the PV energy production, detailed for 4 typical days representative of the seasons of the year.

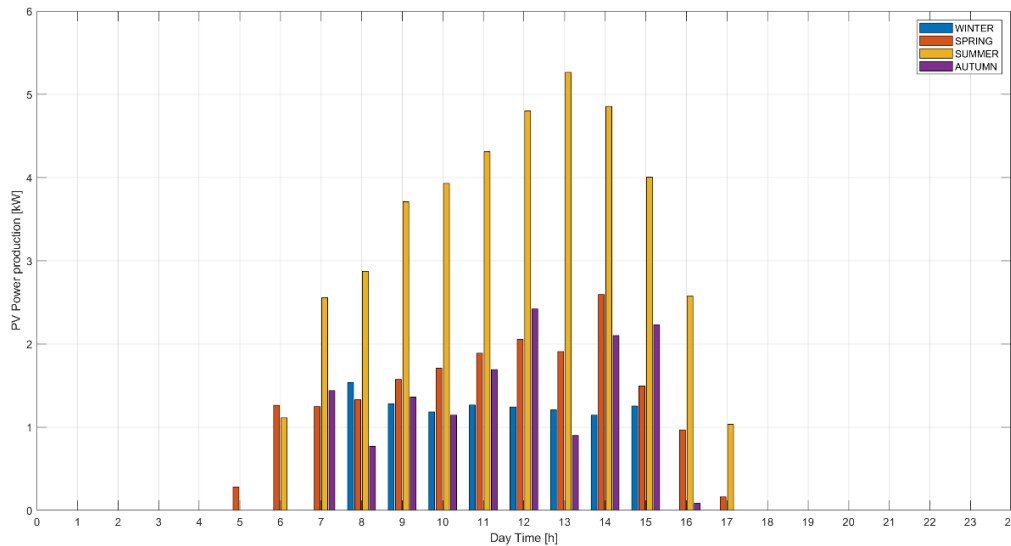

**Figure 6.** Estimation of power that can be produced by a residential PV system during a typical day in different seasons.

By comparing Figures 4 and 6, it can be observed that the PV energy production can only partially fulfill the overall energy demands. Particularly:

- during summer, PV energy production can only partially be used for HVAC systems, thus promoting the self-consumption of the electricity produced;
- during winter, in many hours of the day the energy requirements can be significantly greater than the local availability of electricity produced by PV systems.

The parallel installation of adequate local energy storage systems together with PV plants would therefore be advisable, for solving the above issues. However, most of the PV systems installed to date in Italy are not equipped with storage systems. From this point of view, a vehicle ESS may partially solve the issue, being used instead of a stationary one.

### 2.3. Preliminary Dimensioning, Performance Indicators and Management Algorithms

Once the end users and their electrical behavior are fully defined, it is possible to assume the availability of a vehicle ESSs, temporarily connected to the local network. The following assumptions were made about the ESS, which are representative of a mean-sized EV with a typical daily usage:

- Energy storage capacity equal to 40 kWh.
- Average daily consumption of the vehicle of 12.8 kWh (equivalent to that of a medium-sized EV (weighing approximately 1.5 tons) running daily for 50 km in an urban context (overall energy consumption of 170 Wh/km/ton).
- Maximum power capabilities of the V2H device: 4 kW in both directions (V2H and H2V).

Two indicators were defined to evaluate the operational efficiency of the V2H system:

1. **PV Auto-Consumption Ratio**, representing the fraction of PV production that must not be transferred to the grid, as it finds an available local consumption. This ratio ranges from a minimum "base" value resulting from the local intersection between supply from PV and local energy demand, up to a maximum optimal value of 1, stating that no energy from PV is "sold" to the grid. It can be increased by implementing the V2H charging function of the electric vehicle. Consider that, from this point of view, a benefit can be obtained also through V2H devices simply implementing one-directional functionalities, sometimes referred as V1H (or V1G) devices, which are only able to modulate charging power during charging operations.

2. **Home Auto-Feeding Ratio**, representing the fraction of total domestic consumption (charging of vehicle ESS excluded) that is fed by local systems (through PV systems and/or vehicle storage system in the discharge phase). This ratio ranges from a minimum "base" value resulting from the local intersection between the supply from PV and local energy demand, up to a maximum optimal value of 1, stating that the only electricity which is "bought" form the grid is that needed to charge the Electric vehicle, which may be limited and predictable in time, and stabilized in power, strongly limiting grid problems, and therefore surely and strongly reducing electricity price. In this regard, consider, for example, that most of electricity companies in Italy currently, tend to favor the dynamical predictability of overall energy demand, "sell" electricity for free for 3–4 h a day (in a contractually pre-defined time period, chosen independently on the daytime).

The main objective of the management system should therefore be the optimization of these two operational indicators. For this purpose, a control algorithm for the network connection infrastructure was defined, whose basic behavior is sketched and summarized in Figure 7:

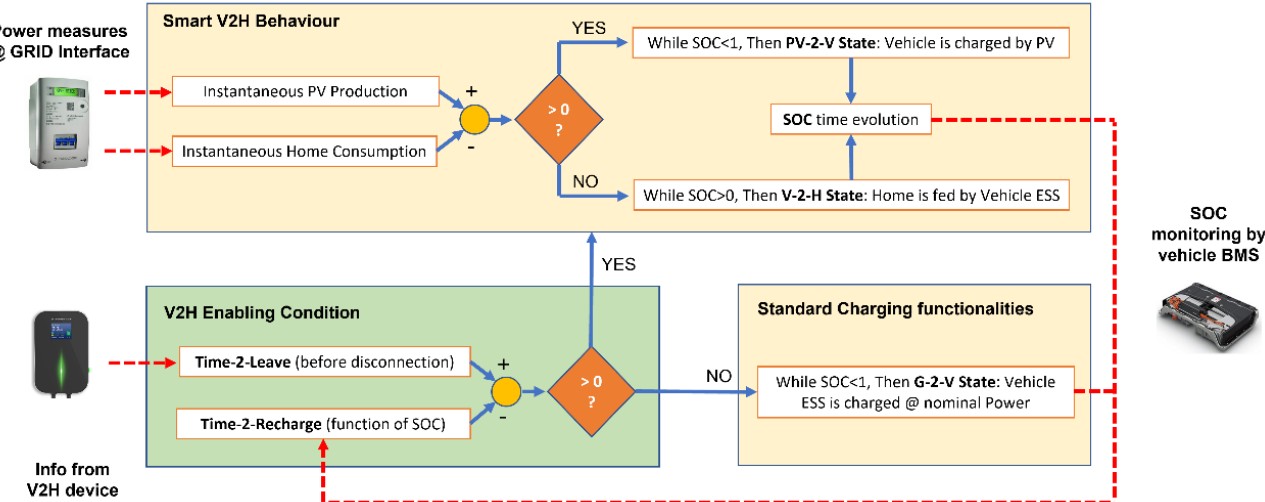

**Figure 7.** Sketch of the proposed algorithm for the optimal energy management of the V2H system.

The main management logic underlying this algorithm is the fulfillment of a pre-condition for which the device can enable V2H functionality (which may be one or bi-directional). This enabling condition is based on the comparison between two timespans, which are instantaneously estimated and monitored by the control algorithm:

- Time-2-Leave (**T2L**), representing the remaining time interval before the planned detachment of the vehicle from the infrastructure. Clearly, an effective calculation of this data presupposes a "collaboration" between the infrastructure and the user. The latter must communicate to the charging infrastructure, introducing the expected time for the next grid disconnection and utilization of the vehicle. Such information must be manually set by the users in the V2H device (see Figure 7), based on the time they are expected to use the vehicle again. If the user anticipates the detachment of the vehicle, it may not be fully charged, while leaving it connected after the expected vehicle pick-up time will inhibit the use in those hours of V2H functionality that could have been implemented, and therefore will not allow the system to further optimize its performance.
- Time-2-Reacharge (**T2R**), representing the time interval for the full charge of the vehicle ESS at nominal infrastructure power. It can be estimated starting from the actual ESS State of Charge (SOC), which, in turn, is monitored by the vehicle BMS and should be read by the V2H controller (e.g., through a standard CAN communication between the two components). T2R is then calculated on the assumption that the charging power is constant at its nominal value.

The algorithm will enable V2H functionality only if **T2L > T2R**. Only in this case, will the system compare the instantaneous values of PV production and home consumption, providing:

- the charge of the vehicle ESS by PV energy, up to the maximum of its nominal charging power, (**PV2V State**), until the ESS reaches full charge;
- the discharge of the ESS to supply residual local loads, up to the maximum of its nominal discharge power, (**V2H State**), until the ESS is completely discharged or reaches a minimum SOC, defined by the user.

If T2L ≤ T2R, the system will enter standard mode charging the vehicle at the nominal rated power (**G2V State**) up to its full charge.

### 2.4. Matlab-Simulink Dynamic Simulation Tool

Starting from all the above-described design assumptions, a software for estimating the expected benefits of appropriately sized V2H systems was developed and realized

in Matlab-Simulink environment with a modular and user-friendly approach. Figure 8 shows the software home screen, consisting of three distinct sections, respectively, devoted to the definition of the dynamical characteristics of: (i) local energy production devices (including PV); (ii) user consumption habitudes (including electric home heating and cooling appliances); (iii) V2H system, including ESS dynamical behavior and its control algorithm modeling.

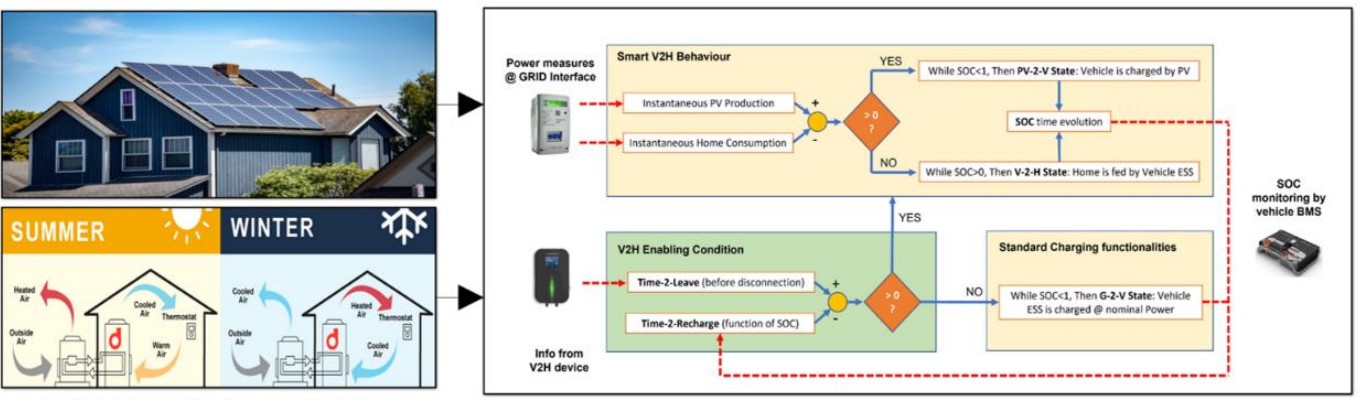

**Figure 8.** Main screen of the developed Matlab-Simulink simulation software.

As underlined, the SW tool was realized to be scalable, modular, and as user-friendly and easy-to-use as possible, with its realization funded by the Italian Ministry of Ecologic Transition (MITE) through its RSE (Electric System Research) funding scheme. All the results produced by this funding scheme, including the present SW tool, are in fact open-source and are made available under request to the authors to be freely used by other research groups for individual and/or cooperating activities.

Within the software, all the functional links between the various elements of the network were defined, following a "blocks-approach" which is typical of the Simulink environment. The energy management algorithm of the V2H system, which was described in the previous section, was also modeled to evaluating its effectiveness in many simulated situations, differing by the type of user and V2H functionality implemented (one or bi-directional). Figure 9 shows the block diagram designed to the implement the described control algorithm.

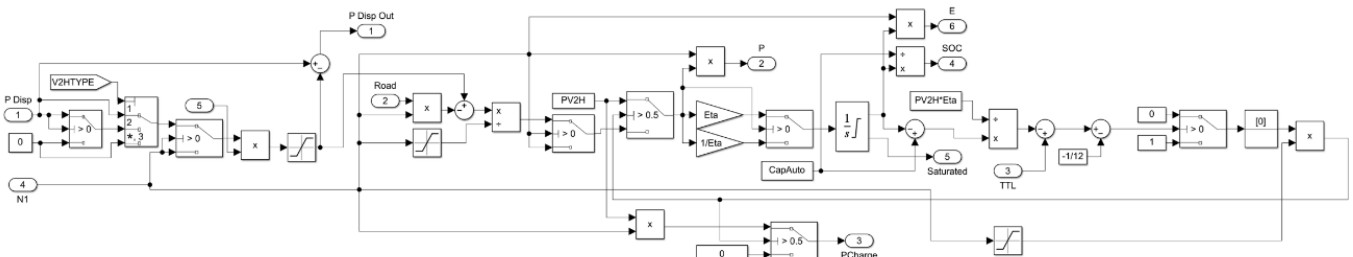

**Figure 9.** Block diagram of the implemented control strategy for the V2H system.

## 3. Results

The developed simulation tool was applied for the various categories of V2H end users, for both one-directional and fully bi-directional functionality. Table 1 summarizes the simulated cases.

**Table 1.** Simulated Cases.

| Case ID | Case Description |
| --- | --- |
| Base Case | V2H function disabled—Office Worker user |
| Case1A—OW 1D | V2H single-directional function—Office Worker user |
| Case1B—OW 2D | V2H bi-directional function—Office Worker user |
| Case2A—SW 1D | V2H single-directional function—Smart Worker user |
| Case2B—SW 2D | V2H bi-directional function—Smart Worker user |
| Case3—MW 2D | V2H bi-directional function—Morning Shift Worker user |
| Case4—AW 2D | V2H bi-directional function—Afternoon Shift Worker user |
| Case5—NW 2D | V2H bi-directional function—Nocturne Shift Worker user |

The performance of V2H systems was evaluated with reference to a Base Case in which V2H functions are not implemented. For all the five types of end users, in fact, the results of the simulations in the Base Case were very similar, as expected. For this reason, in the following, only the results for the **OW** category are here presented and used as reference.

As shown in Table 1, for Cases 1 and 2 (**OW** and **SW** categories) both one and bi-direction V2H functionalities were tested. Results showed that much greater benefits may be obtained through a bi-directional approach. The results here reported about the analyses made for **MW**, **AW**, and **NW**, were therefore limited to the more promising cases implementing fully bi-directional V2H functionalities.

The results of the simulations are here presented in terms of temporal evolution of the main power terms defining the behavior of the various network nodes. More in detail, in each of the following graphs:

- The first row shows the overall electricity production (**PV** thicker curve) and its distribution among the secondary flows: towards domestic users (**PV2H**); towards the vehicle storage system (**PV2V**); sold to the network (**PV2G**).
- The second row shows the overall domestic consumption (**HOME Cons**, thickest curve) and its distribution between two of the three secondary flows: from the network (**G2H**); from the vehicle storage system (**V2H**).
- The third row shows the total electricity withdrawal from the network (**G**, thickest curve) and its distribution among secondary flows: to the vehicle for its recharging (**G2V**); towards the house (**G2H**); from the PV plant being sold to the grid (<0) (**PV2G**).
- The fourth row shows the number of vehicles connected to the infrastructure (1 if connected; 0 if not connected).
- The fifth row shows the evolution of the **SOC** for the vehicle ESS.
- The sixth row shows the ESS power flows. The thickest curve (**ESS**) reports the sum of all instantaneous flows involving the storage system. The other curves show the division between road consumption (**V2Road**, <0), network charging (**G2V**), recharge from PV (**PV2V**), and power supply of households (**V2H**, <0).

All the simulations always considered the entire year of operation of the plant to evaluate its cumulative performance in all possible operating conditions. However, in the following, in order to make the analysis of the results more immediate, only the plots for typical summer and winter weeks are reported. Indeed, both during summer and winter, due to the combination of PV production and relevant consumption, the V2H systems have the potential to demonstrate the greatest benefits, absorbing PV production and/or powering the more energy-intensive loads.

After presenting and discussing the time variation plot for the main variables, for each of the case studies, summary tables are reported relative to the annual integral energy balances at the main nodes of the network and to the evaluated values for the two proposed performance indicators (PV Auto-Consumption Ratio and Home Auto-feeding Ratio).

### 3.1. Base Case Results

Figures 10 and 11 show the results of the simulation for the base case, for a summer and a winter week, respectively.

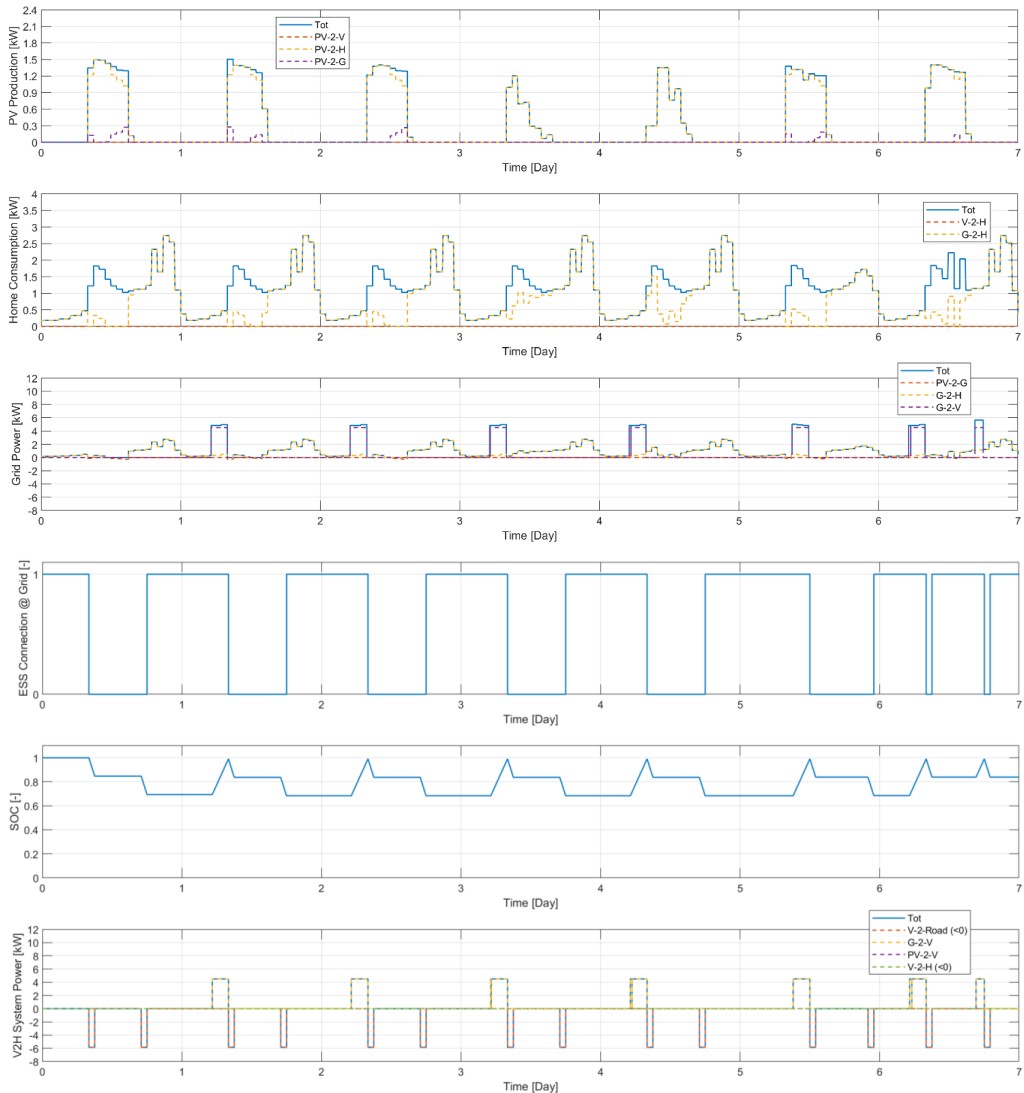

**Figure 10.** Summer week. Base Case results.

Figure 12 shows the annual energy balances for the three main node of the V2H network, namely PV system, home and grid, and the values of the performance indicators.

The values show that self-consumption of energy from PV is about 25%, and feeds about 27% of domestic loads (excluding vehicle charging).

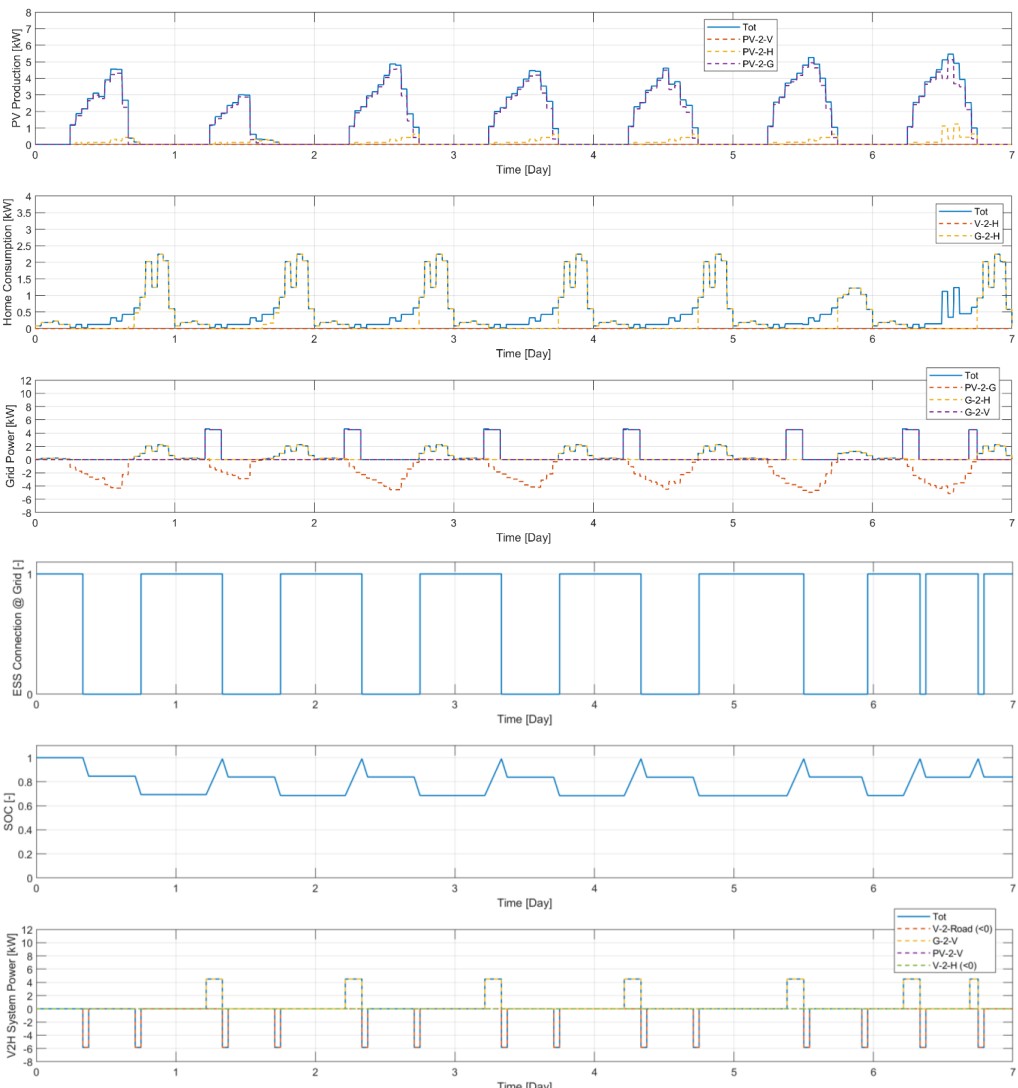

**Figure 11.** Winter week. Base Case results.

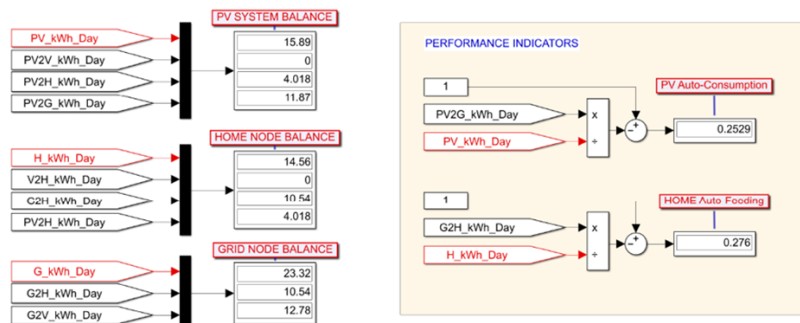

**Figure 12.** Energy balances at the main nodes (**left**) and performance indicators (**right**) for the Base Case.

## 3.2. Case1A—OW 1D Results

In this case, simple one-directional V2H functionalities are enabled. This means that the V2H device only has the possibility to modulate in power the charging over time of vehicle ESS trying to maximize the self-consumption of energy from PV; only one of the performance indicators is therefore expected to grow.

Figure 13 shows the results for a typical summer week, while Figure 14 shows the annual energy balances and the calculated performance indicators.

As highlighted in Figure 13, the Office Worker user keeps the vehicle disconnected from the home network during most of the hours of maximum insolation. This results in only a slight increase in the performance indicators compared to the Base Case. In particular, the self-consumption of PV production only rises to 33%, from 25% in the base case (see Figure 14).

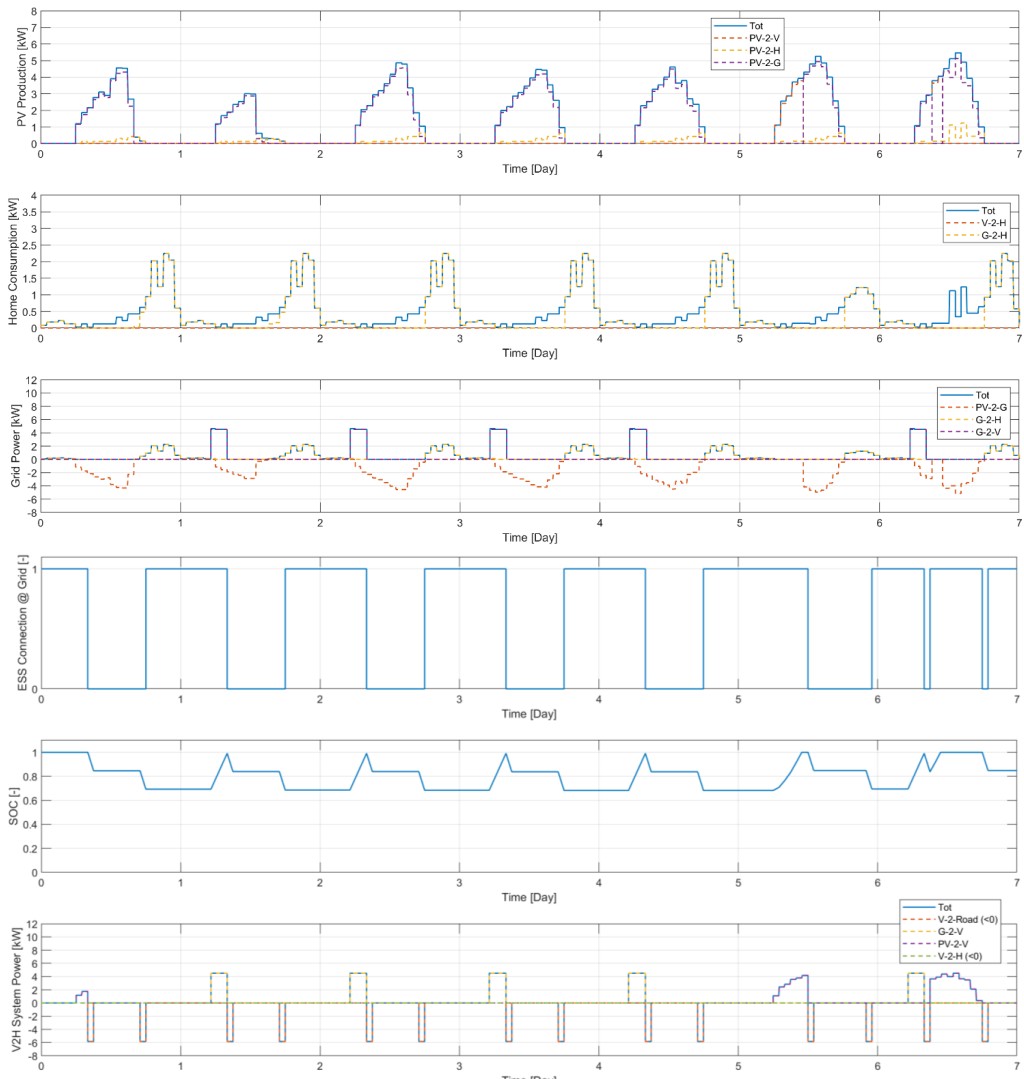

**Figure 13.** Summer week. Case1A—OW 1D results.

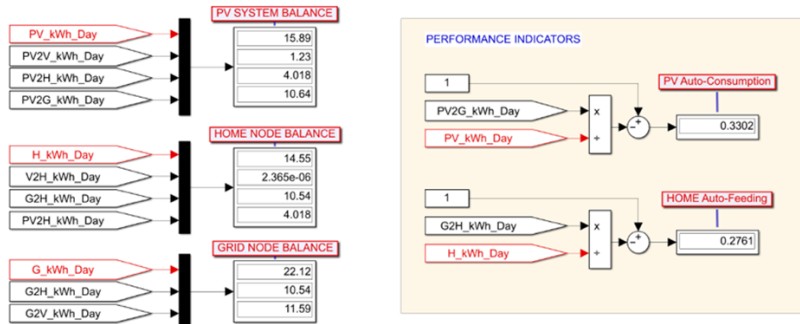

**Figure 14.** Energy balances at the main nodes (**left**) and performance indicators (**right**) for Case1A—OW 1D.

### 3.3. Case1B—OW 2D Results

In this case, the V2H functionality is enabled in the bi-directional mode, also providing for the possibility of powering domestic loads through the vehicle ESS connected to the network. The results are shown in Figures 15 and 16.

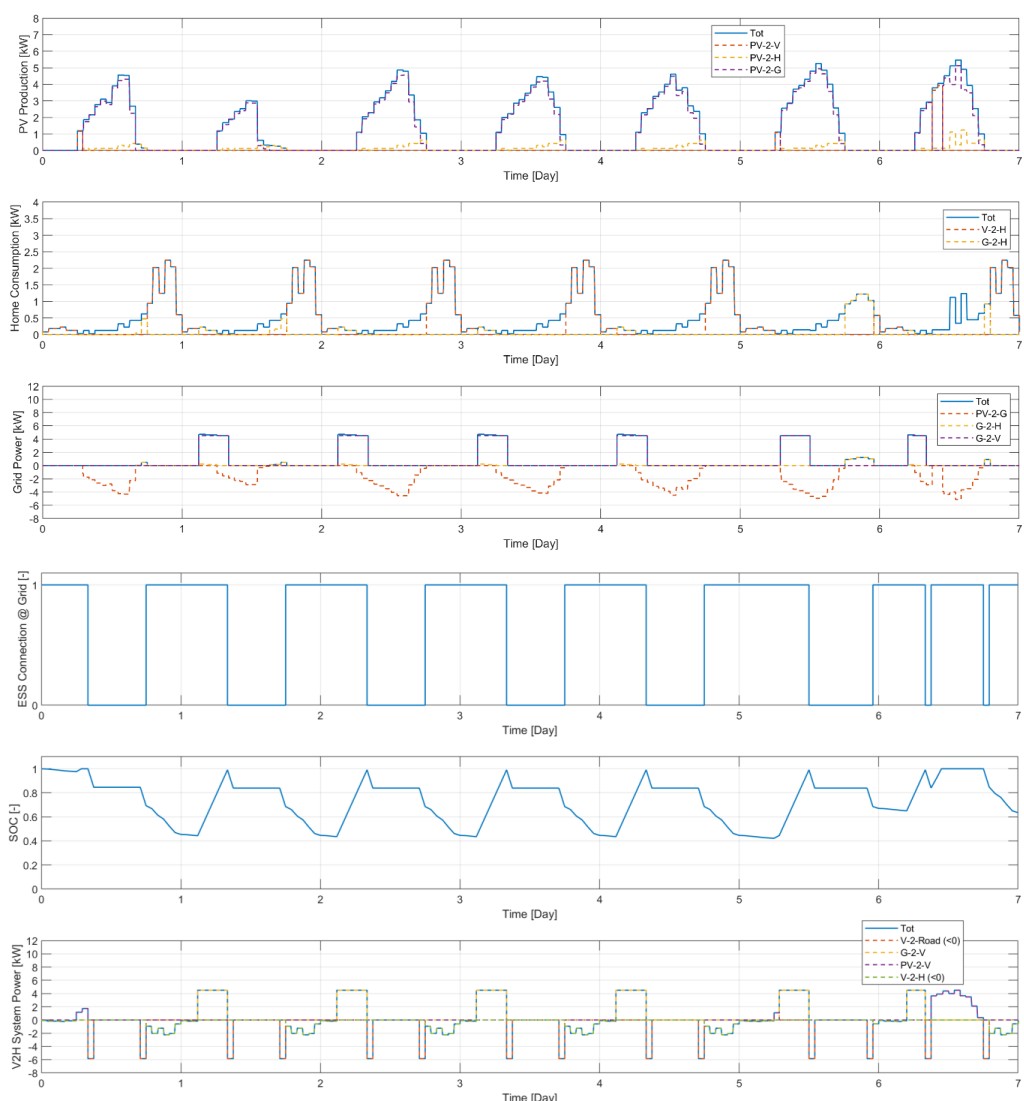

**Figure 15.** Summer week. Case1B—OW 2D results.

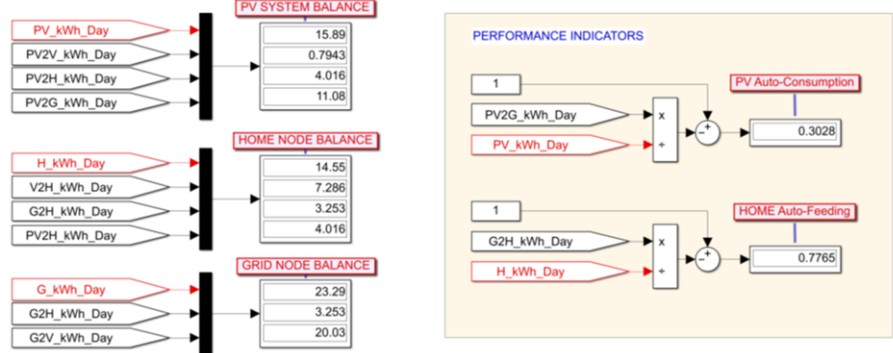

**Figure 16.** Energy balances at the main nodes (**left**) and performance indicators (**right**) for Case1B—OW 2D.

Figure 15 shows how the bi-directional functions allow the feeding of a good part of the domestic loads through the ESS connected to the network, which also has an energy capacity largely sufficient for these purposes (its SOC never falls below 0.4 during the year). However, as highlighted in Figure 16, the self-consumption indicator of the PV is still quite low, even lower than the previous case of mono-directional V2H functions: about 30% against 25% of the basic case and 33% of the mono-directional. Conversely, to the home auto-feeding ratio is greatly increased up to almost 78%, against 27% for the previous cases.

### 3.4. Case2A—SW 1D Results

The Smart Worker user represents, potentially, the category for which the highest performances are expected, given that the ESS remains connected to the grid for most of the day, as shown in Figure 17. Both the graphs in Figures 17 and 18 show what is expected. The much greater coupling of the solar source with the availability of the connected vehicle allows the realization of a good part of the vehicle recharge via solar source, bringing the self-consumption of energy from PV to 55% on an annual basis (compared to 25% of the Base Case).

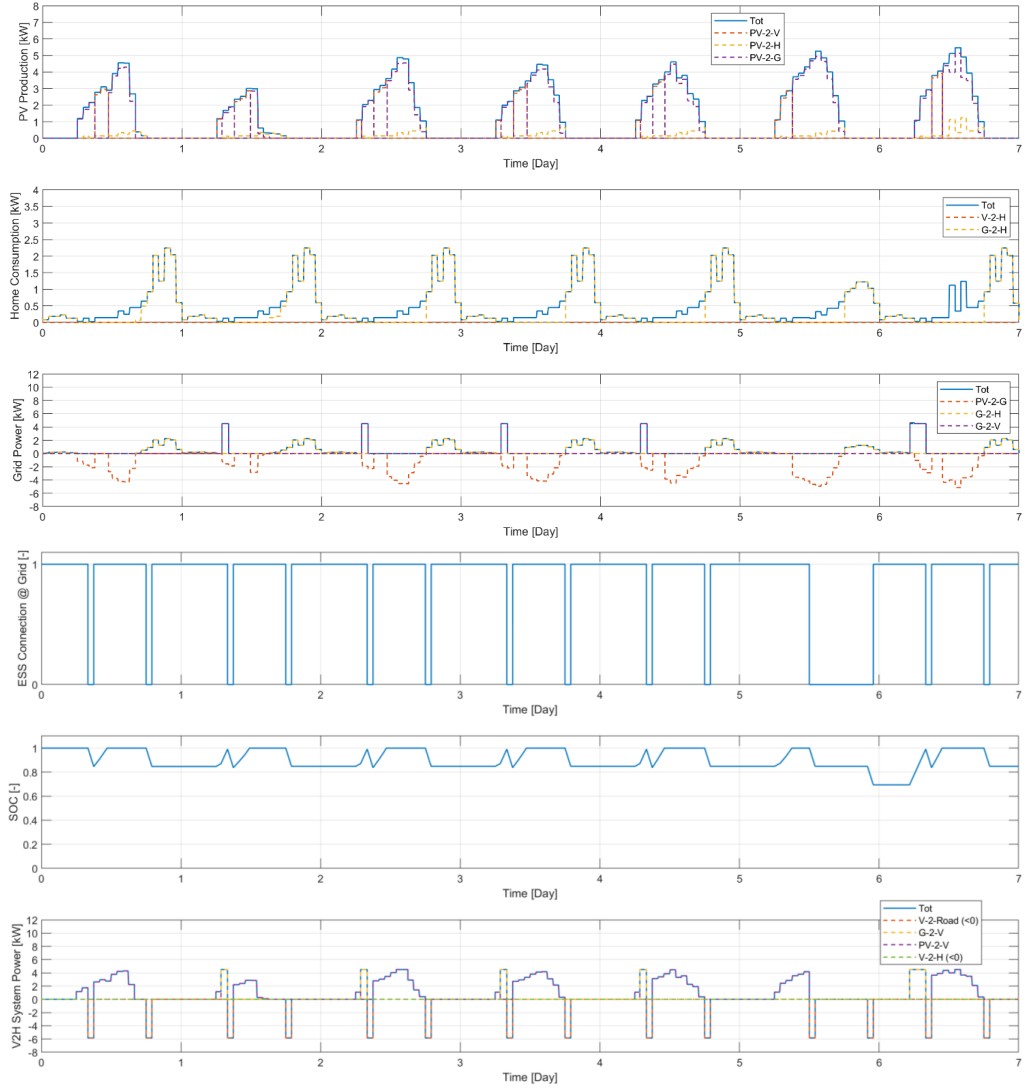

**Figure 17.** Summer week. Case2A—SW 1D results.

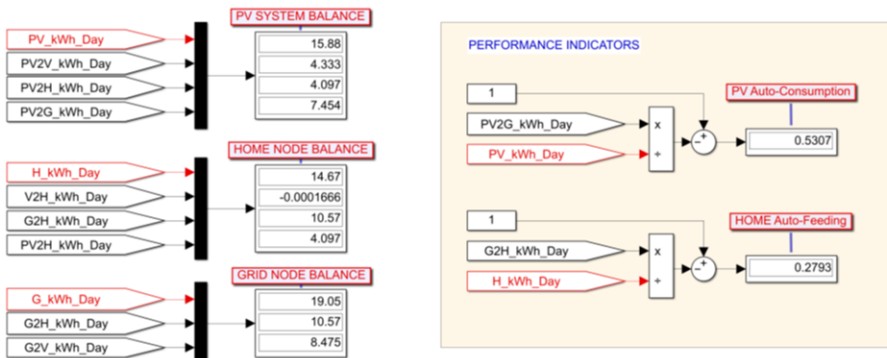

**Figure 18.** Energy balances at the main nodes (**left**) and performance indicators (**right**) for Case2A—
SW 1D.

### 3.5. Case2B—SW 2D Results

In addition to the advantages in terms of self-consumption from PV, typical of this category of user, in the case of bi-directional flows it is also possible to maximize the parameter relating to the self-supply of domestic consumption, as shown in Figures 19 and 20.

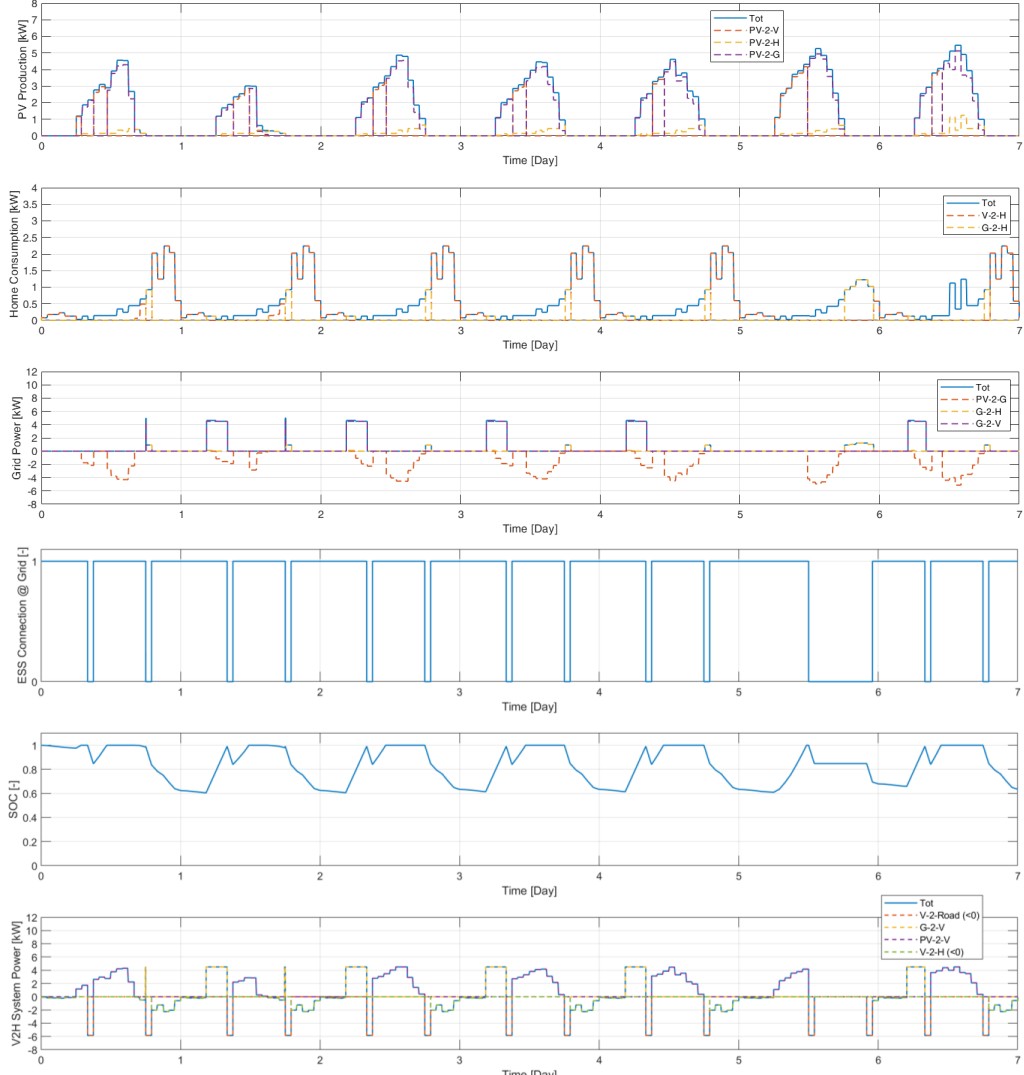

**Figure 19.** Summer week. Case2B—SW 2D results.

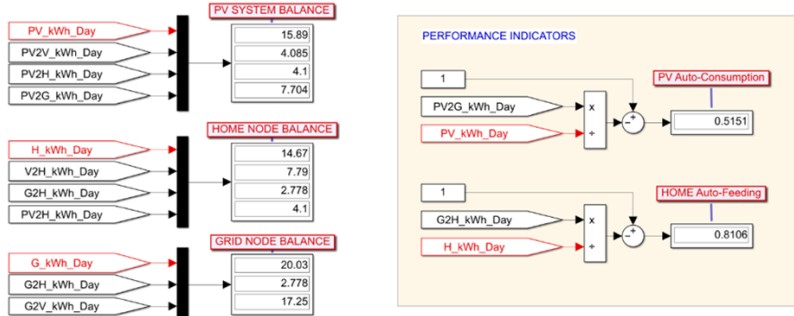

**Figure 20.** Energy balances at the main nodes (**left**) and performance indicators (**right**) for Case2B—SW 2D.

As shown in Figure 20, self-consumption from PV slightly drops compared to Case2A (51% versus 53%), while self-supply of domestic consumption goes from 27% in the Base Case to 81%.

*3.6. Case3—MW 2D Results*

As already discussed, for the sake of brevity, for all shift worker cases (categories **MW**, **AW** and **NW**) the attention will be limited to bi-directional V2H functionalities, and only the overall energy balances and performance indicators results are here presented.

Figure 21 shows the results for the Morning shift Worker (**MW** user category) and bi-directional functionality of the V2H system. The availability of the connection of the ESS starting from the early afternoon allows good V2H functions with performances almost equal to those obtainable for "Smart-Worker" users. In particular, the self-consumption of energy from PV is 41%, and most of the domestic loads can be powered through V2H system (81%).

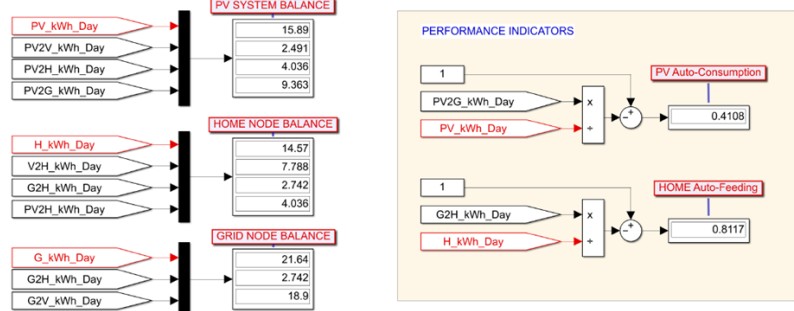

**Figure 21.** Energy balances at the main nodes (**left**) and performance indicators (**right**) for Case3—MW 2D.

*3.7. Case4—AW 2D Results*

Figure 22 shows the results for the Afternoon shift Worker (AW) and bi-directional functionality of the V2H system.

The availability of the connection of the ESS in most of the hours of maximum insolation allows to maximize the self-consumption from PV (which rises to 65% compared to 25% of the base case), but it slightly lowers the possibilities of feeding part of the domestic loads, especially those of heating and air conditioning in the afternoon. The relative performance index drops to 57%, compared to 81% that can be reached by Smart or Morning Workers users.

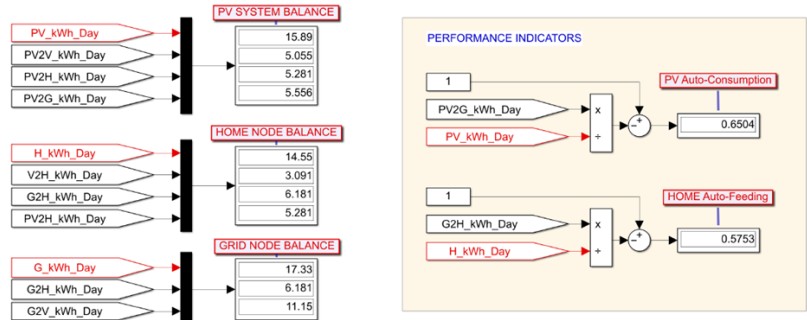

**Figure 22.** Energy balances at the main nodes (**left**) and performance indicators (**right**) for Case4—AW 2D.

*3.8. Case5—NW 2D Results*

Figure 23 shows the results for the Nocturne shift Worker (NW) and bi-directional functionality of the V2H system. The availability of the connection of the ESS in most of the daytime hours allows maximization of both the self-consumption from PV (which rises to 69% compared to 25% of the Base Case), and the possibility of powering many of the domestic loads via V2H, with the home auto-feeding performance index that reaches the 60%.

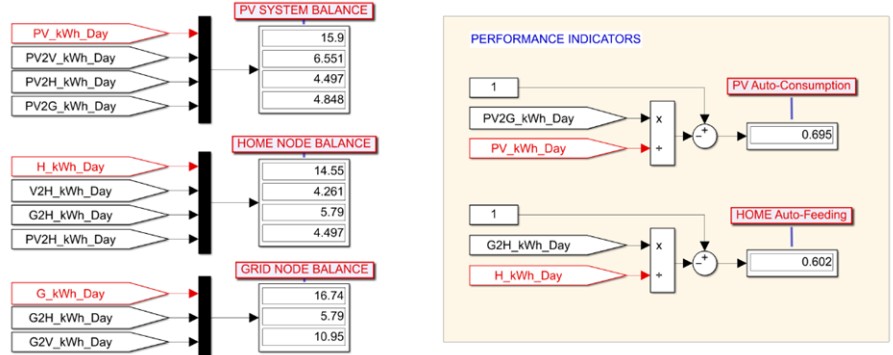

**Figure 23.** Energy balances at the main nodes (**left**) and performance indicators (**right**) for Case5—NW 2D.

Table 2 provides a summary of the main results obtained through the simulations.

**Table 2.** Summary of the main results for all the simulated conditions.

| Case ID | Annual Energy Balances [kWh/Day] | | | | | | | | | | | Performance Indicators | |
| | PV System | | | | Home Node | | | | Grid Node | | | PV Auto-Consumption | Home Auto-Feeding |
| | PV2V | PV2H | PV2G | PV Tot | H2V | H2G | H2PV | H Tot | G2H | G2V | G Tot | | |
| Base Case | 0 | 4.018 | 11.87 | | 0 | 10.54 | 4.018 | **14.56** | 10.54 | 12.78 | **23.32** | 25.29% | 27.60% |
| Case1A—Office Worker 1D | 1.23 | 4.018 | 10.64 | | 0 | 10.54 | 4.018 | **14.56** | 10.54 | 11.59 | **22.12** | 33.02% | 27.61% |
| Case1B—Office Worker 2D | 0.794 | 4.016 | 11.08 | | 7.286 | 3.253 | 4.016 | **14.56** | 3.253 | 20.03 | **23.29** | 30.28% | 77.65% |
| Case2A—Smart Worker 1D | 4.333 | 4.097 | 7.454 | | 0 | 10.57 | 4.097 | **14.67** | 10.57 | 8.475 | **19.05** | 53.07% | 27.93% |
| Case2B—Smart Worker 2D | 4.085 | 4.1 | 7.704 | **15.89** | 7.79 | 2.778 | 4.1 | **14.67** | 2.778 | 17.25 | **20.03** | 51.51% | 81.06% |
| Case3—Morning Worker 2D | 2.491 | 4.036 | 9.363 | | 7.788 | 2.742 | 4.036 | **14.57** | 2.742 | 18.9 | **21.64** | 41.08% | 81.17% |
| Case4—Afternoon Worker 2D | 5.055 | 5.281 | 5.556 | | 3.091 | 6.181 | 5.281 | **14.55** | 6.181 | 11.15 | **17.33** | 65.04% | 57.53% |
| Case5—Nocturne Worker 2D | 6.551 | 4.497 | 4.848 | | 4.261 | 5.79 | 4.497 | **14.55** | 5.79 | 10.95 | **16.74** | 69.50% | 60.20% |

## 4. Discussion

In this study the authors presented a mathematical dynamic tool for evaluating the benefits achievable using intelligent V2H charging systems for electric cars. The SW tool was realized to be scalable, modular, and as user-friendly and easy-to use as possible, with its realization funded by the Italian Ministry of University and Scientific Research (MUR) through its RSE (Electric System Research) funding scheme. All the results produced by this funding scheme, including the present SW tool, are in fact open-source and are made available under request to the authors to be freely used by other research groups for individual and/or cooperating activities.

The SW tool was developed within a broader collaboration between the University of L'Aquila, the Cassino University and the Italian National Agency for New Technologies, Energy and Sustainable Economic Development (ENEA), which includes the realization and testing of a wireless V2H prototype.

The approach proposed was chosen to be replicable and modular also permitting its generalization and application to small/medium building contexts, in so-called V2B. This activity is currently under development and will be soon published in a further paper by the same authors.

The proposed SW tool was first used to simulate several possible users and contexts identifying the most promising conditions for V2H devices. An effective energy management algorithm aimed at optimizing network performances was defined and modeled as well. Lastly, two different indicators were introduced to evaluate the operational efficiency of the V2H system: the PV Auto-Consumption Ratio (estimating obtainable self-consumption of energy produced from renewable sources); and the Home Auto-Feeding Ratio (estimating fraction of total domestic consumption that can be locally fed by PV systems and/or vehicle ESS discharging).

The developed tool can be easily implemented on real-time machines for managing V2H systems, not requiring particular information from the vehicle's ESS BMS.

The software was tested in both one or bi-directional V2H function modes and for the different types of users.

The results showed that it is possible to obtain significant overall benefits for all types of both V2H function mode and users, compared to a base case in which the V2H strategy is not implemented. However, the combination of a bi-directional approach with users who can remain connected to the grid for most of the hours in which a solar source is available (smart and morning shift workers) allowed obtainment of the highest performances both in terms of Home Auto-Feeding ratio and PV Auto-Consumption index.

The application of the developed simulation tool can be leveraged for the optimal design and energy management of V2H systems in a single-user residential context.

Moreover, the same approach can be applied for small/medium building contexts, namely for V2B applications. In fact, in the near future authors will be able to demonstrate in a further work the applicability of the same modeling and controlling approach to condominium networks individuating the most appealing application conditions and the relative expected performances of the implementable V2B systems.

**Author Contributions:** Conceptualization, C.V., S.R., F.D., A.D.V. and M.A.; methodology, C.V., S.R., F.D., A.D.V. and M.A.; software, C.V., S.R., F.D., A.D.V. and M.A.; validation, C.V., S.R., F.D., A.D.V. and M.A.; writing—original draft preparation, C.V., S.R., F.D., A.D.V. and M.A.; writing—review and editing, C.V., S.R., F.D., A.D.V. and M.A.; visualization, C.V., S.R., F.D., A.D.V. and M.A.; supervision, C.V; project administration, C.V., A.D.V. and M.A.; funding acquisition, C.V., A.D.V. and M.A. All authors have read and agreed to the published version of the manuscript.

**Funding:** This research has been funded by Italian Ministry of Ecologic Transition (MITE) through its RSE (Electric System Research) Funding scheme, among the Key Actions for Energy efficiency improvement in electro-mobility.

**Institutional Review Board Statement:** Not applicable.

**Informed Consent Statement:** Not applicable.

**Data Availability Statement:** Not applicable.

**Conflicts of Interest:** The authors declare no conflict of interest.

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
