# Peer review of "An Energy-Based Assessment of Expected Benefits for V2H Charging Systems through a Dedicated Dynamic Simulation and Optimization Tool"

_wevj, doi:10.3390/wevj13060099_

Round 1

Reviewer 1 Report

In the article authors developed a mathematical tool able to simulate V2H systems. They also synthetized and tested an effective energy optimization algorithm which can be easily implemented on the devices.

The article is written and edited correctly. The obtained results are promising for future implementations. The article requires minor corrections, which are indicated below.

1.      In line 168 there should be Figure 4 instead of Figure 2

2.      Improve the quality of figures 3, 4, 7, 10, 11.

3.      In my opinion, the presentation of Figure 8 adds nothing to the article and it can be removed.

Authors should refer to the following critical remarks:

1.      The algorithm proposed in the article, presented in Fig. 7, is very simple and consists in calculating two times, T2R, T2L. Please describe more information on how these times are calculated. How are they estimated and what are the limitations for the owner of the electric vehicle related to the implementation of the algorithm and if he has to interrupt the algorithm.

2.      A detailed description of the algorithm presented in figure 9 should be added. This is the focal point of the article, and there is nothing about how it works.

3.      With such illegible drawings showing the test results, it is very difficult to analyze the test results.

Reviewer 2 Report

In this paper, the authors claimed of proposing an effective energy optimization algorithm and a mathematical tool for simulation of V2H system. However, the strategy proposed by the authors is very basic and many methodologies like this have been presented by other researchers during early exploration of this research topic. Furthermore, the authors should address the following comments:

1. The authors should include some key results of their proposed methodology in the abstract section in quantitative form.

2. In line 69, replace word "form" with "from" in the sentence "But problems arise also form".

3. In line 76, "with the" has been written twice at the end of the sentence. Please correct.

4. There are many other typo mistakes in the manuscript. Please carefully read the paper and correct them.

5. The literature review section of the paper is very weak. Include some recent research studies related to the topic in the introduction section.

6. Considering the recent research studies, please mention the research gap for the proposed method.

7. Clearly state the contributions of the proposed technique.

8. The methodology developed in the manuscript is trivial. Therefore, considering the recent research literature, the novelty of the proposed scheme is negligible.

9. The authors have mentioned that they synthesized an effective energy optimization algorithm in this paper. However, there is no objective function considered in the paper and the authors have proposed the methodology using simple decision logics.

10. The authors should include "Conclusion" section in the manuscript.

11. English language used in the paper requires extensive editing.

Reviewer 3 Report

The authors focus their study on the vehicle to home systems by developing a corresponding mathematical tool in order to perform an optimization of the energy and economic management within the vehicle to home systems. The authors have provided a detailed testing of the proposed optimization algorithm regarding the effective energy optimization and they have quantified its benefits. The manuscript is overall well written and the authors have well thought out their main contributions. The provided theoretical analysis is concrete, complete, and correct and the authors have provided all the intermediate steps in order to enable the reader to easily follow it. Furthermore, the provided numerical results are also rich in order to quantify that drawbacks and benefits of the proposed approach. The authors are encouraged to consider the following suggestions provided by the reviewer in order to improve the scientific depth of their manuscript, as well as they need to address the following comments in order to improve the quality of presentation of their manuscript. Initially, in Section 1, the authors need to improve the provided related work by using more summative language in order to better identify the research contributions that have already been performed in the literature and the research gap that the authors tried to address. There are several recent approaches that exploit network economics techniques that have been introduced in their literature, such as Contract-Theoretic Demand Response Management in Smart Grid Systems, doi: 10.1109/ACCESS.2020.3030195, deal with the implementation of smart grid paradigm towards enabling the effective energy optimization. The authors need to improve the provided related work in order to capture the most recent state of the art. In section 2, the authors need to include an additional subsection providing the computational analysis of the proposed framework and clarifying if it can be implemented in a real time or even close to real time manner. Based on the previous comment, in Section 3, the authors need to provide either some qualitative discussion or some indicative quantitative results in order to capture the computational complexity of the proposed framework in order to be implemented in a realistic vehicles to home system. Finally, the overall manuscript needs to be checked for typos, syntax, and grammar errors in order to improve the quality of its presentation.

Reviewer 4 Report

The paper "An energetic assessment of expected benefits for V2H charging systems through a dedicated dynamic simulation and optimization tool" with Manuscript ID: wevj-1754681 is recommended for minor revision

STRENGTHS:

The paper is well organised.

The paper has good flow and suits WEVJ.

The paper has good references but it needs to be improved. Add more details like DOI.

The paper has good introduction.

The paper has good results and conclusion.

The authors show good knowledge of the subject area.

WEAKNESSES:

1. The paper is a technical paper and some mathematical models used in the study like in Figure 9 should be presented. Add the mathematical equations for them

2. The paper requires more understanding so readers and other EV researchers can replicate these.

3. The paper requires minor English language editing and some proof reading.

4. The title uses "energetic assessment " which means "powerful assessment"  but I believe the authors mean to use "energy-based assessment ", so it should be revised.

Round 2

Reviewer 1 Report

All my suggestion has been taken into account and indicated problems have been solved. Additional comments, which have been added on a request of the other reviewers also improve the scientific level of the paper. Therefore, in my opinion, the paper in the current version is suggested to be accepted in WEVJ  Journal.

Reviewer 2 Report

The authors have satisfactorily addressed all the comments.

Reviewer 3 Report

The authors have addressed the reviewers comments in detail.